Biocontrol potential of endophytic bacterium Bacillus altitudinis GS-16 against tea anthracnose caused by Colletotrichum gloeosporioides

Wu Youzhen 1 2
Tan Yumei 2
Peng Qiuju 1
Xiao Yang 3
Xie Jiaofu 4
Li Zhu 1 2 zhuliluck@163.com
Ding Haixia 5 hxding@gzu.edu.cn
Pan Hang 1
Wei Longfeng 1
1 Key Laboratory of Plant Resource Conservation and Germplasm Innovation in Mountainous Region (Ministry of Education), College of Life Sciences/Institute of Agro-bioengineering, Guizhou University , Guiyang, Guizhou Province , China
2 Guizhou Key Laboratory of Agricultural Biotechnology, Guizhou Academy of Agricultural Sciences, Institute of Biotechnology , Guiyang, Guizhou Province , China
3 Institution of Supervision and Inspection Product Quality of Guizhou Province , Guiyang, Guizhou Province , China
4 Guiyang No. 1 High School , Guiyang, Guizhou Province , China
5 Department of Plant Pathology, College of Agriculture, Guizhou University , Guiyang, Guizhou Province , China
Yasmin Humaira
Electronic publication date: 2024 Jan 9
Publication date: 2024
Volume: 12
Electronic Location ID: e16761
Received 2023 Aug 17; Accepted 2023 Dec 13
Copyright: © 2024 Wu et al.
Copyright year: 2024
Copyright holder: Wu et al.
License: This is an open access article distributed under the terms of the Creative Commons Attribution License, which permits unrestricted use, distribution, reproduction and adaptation in any medium and for any purpose provided that it is properly attributed. For attribution, the original author(s), title, publication source (PeerJ) and either DOI or URL of the article must be cited.
License URL: https://creativecommons.org/licenses/by/4.0/

Keywords: Endophytic bacterium, Camellia sinensis, Colletotrichum gloeosporioides, Bacillus altitudinis, Biocontrol efficacy

Funding: Guizhou Province High-level Innovative Talent Project Qiankehe Platform Talent-GCC[2022]027-1 Science and Technology Project of Guizhou Province [2021]193 Central government guidance for local science and technology development projects for Guizhou province [2023]027 Modern industrial technology system for Chinese medicinal materials in Guizhou Province This work was supported by the Guizhou Province High-level Innovative Talent Project (Qiankehe Platform Talent-GCC[2022]027-1), the Science and Technology Project of Guizhou Province (grant number [2021]193), the Central government guidance for local science and technology development projects for Guizhou province ([2023]027), and the Modern industrial technology system for Chinese medicinal materials in Guizhou Province. The funders had no role in study design, data collection and analysis, decision to publish, or preparation of the manuscript.

==============================
Background

As one of the main pathogens causing tea anthracnose disease, Colletotrichum gloeosporioides has brought immeasurable impact on the sustainable development of agriculture. Given the adverse effects of chemical pesticides to the environment and human health, biological control has been a focus of the research on this pathogen. Bacillus altitudinis GS-16, which was isolated from healthy tea leaves, had exhibited strong antagonistic activity against tea anthracnose disease.

Methods

The antifungal mechanism of the endophytic bacterium GS-16 against C. gloeosporioides 1-F was determined by dual-culture assays, pot experiments, cell membrane permeability, cellular contents, cell metabolism, and the activities of the key defense enzymes.

Results

We investigated the possible mechanism of strain GS-16 inhibiting 1-F. In vitro, the dual-culture assays revealed that strain GS-16 had significant antagonistic activity (92.03%) against 1-F and broad-spectrum antifungal activity in all tested plant pathogens. In pot experiments, the disease index decreased to 6.12 after treatment with GS-16, indicating that strain GS-16 had a good biocontrol effect against tea anthracnose disease (89.06%). When the PE extract of GS-16 treated mycelial of 1-F, the mycelial appeared deformities, distortions, and swelling by SEM observations. Besides that, compared with the negative control, the contents of nucleic acids, protein, and total soluble sugar of GS-16 group were increased significantly, indicating that the PE extract of GS-16 could cause damage to integrity of 1-F. We also found that GS-16 obviously destroyed cellular metabolism and the normal synthesis of cellular contents. Additionally, treatment with GS-16 induced plant resistance by increasing the activities of the key defense enzymes PPO, SOD, CAT, PAL, and POD.

Conclusions

We concluded that GS-16 could damage cell permeability and integrity, destroy the normal synthesis of cellular contents, and induce plant resistance, which contributed to its antagonistic activity. These findings indicated that strain GS-16 could be used as an efficient microorganism for tea anthracnose disease caused by C. gloeosporioides.

Introduction

Tea (Camellia sinensis (L.) O. Kuntze) is one of the most important cash crops and its valuable metabolites are beneficial for human health (Pan et al., 2022a). However, the occurrence of tea anthracnose disease caused by Colletotrichum gloeosporioides has been reported in the main tea-producing areas. Tea anthracnose disease is a destructive plant disease that can result in substantial losses in yield and reductions in quality (Shi et al., 2018). Tea is infected through a wound in the natural pore by the conidia of Colletotrichum spp. (Münch et al., 2008), which can lead to infection in the buds, fruits, and leaves of tea plants can be infected by Colletotrichum spp., causing a 20–40% fruit drop and up to 40% seed loss (Zhu et al., 2015).

The main measures for controlling tea anthracnose disease include prevention and treatment (Chen et al., 2022). Breeding and cultivating disease-resistant plants is an effective approach to control tea anthracnose disease (Savchenko, 2017). It was previously found that Camellia oleifera Abel var. Huizhou-xiaohong, Camellia yuhsienensis and Camellia octopetala expressed resistance to Colletotrichum spp. (Duan et al., 2005). Nevertheless, tea plant breeding is difficult because of the long process and low breeding efficacy (Wang et al., 2016). Control of tea anthracnose disease is mainly achieved by chemical treatment. The Bordeaux mixture can effectively control tea anthracnose disease in the early stage, and chlorothalonil, carbendazim, and thiophanate methyl can effectively control tea anthracnose disease in the late stage (Yu, 2019). However, the excessive use of synthetic fungicides not only results in the emergence of pathogens resistant to synthetic fungicides but also has negative effects on the environment and human health (Holtappels et al., 2021). In contrast, biocontrol agents (BCAs) have several benefits, including being effective, simple to apply, and ecologically beneficial (Kim et al., 2016). Therefore, biological control has become a major research focus.

Currently, endophytes are an important source of BCAs (Zhang et al., 2022), and have attracted considerable attention from researchers (Ji et al., 2008). In recent years, endophytic microorganisms of Bacillus spp. have shown promise as BCAs for the control of phytopathogens because of their strong effects, broad-spectrum antimicrobial properties and strong safety profile (Kaushal, Kumar & Kaushal, 2017; Cheng et al., 2019). For instance, Zheng et al. (2021) isolated 31 endophytic bacteria from Fraxinus hupehensis, and Bacillus velezensis D61-A showed strong antagonistic activity against eight different pathogenic fungi. It is known that Bacillus spp. act through multiple mechanisms, such as producing various secondary metabolites, hydrolytic enzymes, and inducing plant defense responses via systemic resistance to pathogens (Lopes et al., 2018). Bacillus spp. exhibiting activity against tea anthracnose disease have been reported. Zhou et al. (2008) found that Bacillus subtilis Y13, which isolated from healthy tea leaves, had an inhibition rate of 88.5% against Colletotrichum spp. and Shang et al. (2021) observed that B. velezensis HBMC–B05 was obtained from healthy C. oleifera leaves, which showed a strong inhibition rate (88.5%) against Colletotrichum spp.

Despite these studies, few biocontrol treatments for tea anthracnose disease have been commercialized, possibly due to low efficacy (Zheng et al., 2021). Anthracnose remains a challenge in the tea industry, despite significant advancements in research on the control of this disease. Therefore, the aim of this research is to explore more effective endophytic bacteria for controlling tea anthracnose disease. In this study, the following were performed: (1) we assessed the biocontrol efficacy of Bacillus altitudinis GS-16 in vitro and in vivo and (2) we analyzed the antagonistic mechanisms of the B. altitudinis GS-16 from the aspects of mycelial growth, cell membrane integrity and permeability, cellular metabolism and the induction of systemic resistance.

Materials and Methods

Strains and conditions

In this study, 2-year-old tea seedlings (C. sinensis cv. Fuding Dabaicha) were obtained from Guiyang City, Guizhou Province, China. A total of nine common agricultural phytopathogenic fungi were used to assess the antifungal activity of B. altitudinis GS-16, and C. gloeosporioides 1-F (tea anthracnose disease, Chen et al., 2022), Colletotrichum camelliae Tj-26 (brown blight disease, Kong et al., 2023), Plectosphaerella cucumerina Z-14 (strawberry wilt, Yang et al., 2023a), Fusarium dimerum X1-19 (fusarium rot pathogen, Hashem et al., 2010), Fusarium equiseti B-3-1 (soybean root rot, Chang et al., 2018), Fusarium graminearum Z-16 (soybean root rot, Chang et al., 2018), Fusarium oxysporum X1-16 (banana fusarium wilt pathogen, Magdama et al., 2020; Maryani et al., 2019), and Alternaria alternata A-3 (pear black spot pathogen, Tanahashi, Nakano & Akamatsu, 2018) were isolated, identified and preserved by the Biotechnology Laboratory, College of Life Sciences, Guizhou University, Guizhou Province, China. Another fungus tested was Sclerotium rolfsii R-67 (pepper southern blight, Fery & Dukes, 2005), which was purchased from the Agricultural Culture Collection of China. All fungal strains were cultured in potato dextrose agar (PDA) medium at 28 °C (Yang et al., 2023b).

B. altitudinis GS-16 was isolated from healthy leaves of C. sinensis in Guizhou Province, China, and preserved by the China General Microbiological Culture Collection Center (CGMCC) under the accession number 24353. B. altitudinis GS-16 was cultured in Luria-Bertani (LB) at 30 °C.

Determination of the antagonistic activity in vitro

The antagonistic activity in vitro was determined according to the method described by Jiang et al. (2018) with some modifications. A mycelial disc of 1-F (diameter 5 mm) cultured for 5 days was moved to the center of a PDA plate, and the strain GS-16 was placed 1.5 cm from the sides of the mycelial disc for incubation at 28 °C for 7 days. Plates without bacteria were used as controls, and the experiments were performed in triplicate. The diameters of the pathogen colonies were measured (in mm), and the inhibition rates were determined with the following formula (Chen et al., 2019):

Inhibitionrate(%)=(C−T)/C×100

where C represents the colony diameter of 1-F in the control and T represents the colony diameter in the treatment group.

One single colony of GS-16 was cultured in LB broth medium with shaking at 160 rpm for 48 h at 30 °C. The bacterial suspension was collected and centrifuged at 10,000 g/min for 20 min at 4 °C. Then, the supernatant was filtered through a 0.22 µm microporous filter to obtain the bacterial culture filtrates (BCF) of GS-16 (Zheng et al., 2021). Inhibition rates were measured after treatment at 28 °C for 7 days based on the ratio of BCF to PDA medium (1: 5, v: v) (Cruz-Martín et al., 2013).

Pot experiment

GS-16 was grown in LB broth at 30 °C with shaking at 200 rpm for 48 h and diluted with sterile water to a final concentration of 1 × 108 cfu/mL. The 1-F was cultured on PDA at 28 °C for 7 days. Then, the mycelia of 1-F were collected and suspended in sterile water, and then filtered through three layers of gauze to obtain a spore suspension. The concentration of the spore suspension was adjusted to 1 × 106 spores/mL using sterile water for further studies. 2-year-old seedlings were randomly divided into three groups (three pots/group) and treated (spraying 5 mL/plant) as follows: (1) sterile water, (2) 1-F spore suspension (1 × 106 spores/mL), and (3) bacterial suspensions of GS-16 (1 × 108 cfu/mL) followed by treatment with the 1-F spore suspension (1 × 106 spores/mL) 24 h later. Then, the tea seedlings were placed in transparent plastic bags to prevent pathogen spread and watered until the soil was wet every three days. Seedlings were cultured on a 12 h light/dark photoperiod (temperature of 25 °C, humidity of 85%). On the 14th day after treatment, the disease incidence under each treatment was calculated as the percentage of diseased plants. The severity of the disease was scored on a 0–5 scale (Yang & Hartman, 2015), and the disease index and control efficacy were determined with the following equations (Luo et al., 2015):

Diseaseindex=∑Numberofdiseasedplantswiththatgrade×DiseasegradeTotalnumberofplantsinvestigated×Highestdiseasegrade×100

Relativebiocontrolefficacy(%)=Diseaseindexofcontrol−DiseaseindexoftheantagonistDiseaseindexofthecontrol×100

Inhibition spectrum of GS-16

Eight different agricultural pathogens were used to assess the antifungal spectrum of GS-16: C. camelliae Tj-26, P. cucumerina Z-14, F. dimerum X1-19, F. equiseti B-3-1, F. graminearum Z-16, F. oxysporum X1-16, A. alternata A-3, and S. rolfsii R-67. The inhibition rates were determined as described in “Determination of the antagonistic activity in vitro”. Each treatment had three replicates, and plates without strain GS-16 were used as controls.

Determination of the EC50

An equal volume of petroleum ether (PE) was added to the BCF of the GS-16 strain, and the organic phase was collected after three extractions. A rotary evaporator was used to evaporate the organic solvent, after which the residue was dried. The product was dissolved in 6% dimethyl sulfoxide (DMSO) to prepare a 10 mg/mL stock solution, filtered through a 0.22 µm microporous filter, and stored at 4 °C for further studies. Plugs of 1-F (5 mm in diameter) were moved to the center of the PDA containing the PE extract of GS-16 (at 1, 0.5, 0.25, 0.125, and 0.0625 mg/mL) or serial 2-fold dilutions of mancozeb (concentrations of 0.5, 0.25, 0.125, 0.0625, 0.03125 mg/mL) as a positive control (Gu et al., 2017). All plates were incubated at 28 °C for 7 days. DMSO (6%) was used as the negative control. Three replicates were maintained in each treatment group. The diameters of the colonies were measured according to Mu et al. (2017). The EC50 values of the PE extract of GS-16 and mancozeb were calculated with GraphPad Prism 9 software.

Effects of the PE extract of GS-16 on the mycelial morphology and cell walls of 1-F

The mycelial morphology of 1-F was observed by scanning electron microscopy (SEM). According to Pepin (1974), the PE extract of GS-16 and mancozeb were mixed with PDA medium at the EC50 concentration, and the mixture was used to culture 1-F at 28 °C for 7 days as described by Cruz-Martín et al. (2013). SEM (Evo Maio Zeiss, Germany) observations were performed following the method of Guo et al. (2017). Mancozeb was used as the positive control, and sterile water was used as the negative control.

The effects of GS-16 on the cell walls of 1-F were determined by a plate-based assay. The plate-based assay for proteinase production was performed using skimmed milk agar medium following the method of Jangir et al. (2018). After 3 days at 28 °C, the development of clear zones around the colonies on the plates was recorded. Cellulase activity was measured using carboxymethyl cellulose (CMC) agar (Zheng et al., 2011). The GS-16 culture was inoculated by spotting on plates containing CMC as the sole source of carbon and incubated at 28 °C for 5 days.

Preliminary investigation of the inhibition mechanism of GS-16

Effects of the PE extract of GS-16 on the cell membrane permeability and cellular contents of 1-F

Five agar discs of 1-F were transferred into 100 mL of potato dextrose broth (PDB) medium and shaken at 180 rpm for 5 days at 28 °C. Then, the PE extract of GS-16 and mancozeb were added to the cultures at the EC50 (final concentration). After the cultures were allowed to stand for 0, 4, 8, 12, 16, 20, 24, 28 and 32 h, 5 mL aliquots were removed and centrifuged (10,000 × g, 10 min). The relative electrical conductivity of each supernatant was determined by a conductivity meter (DDB303A, China) (Song et al., 2016). Mancozeb was used as the positive control, and sterile water was used as the negative control. Each treatment had three replicates. The leakage of soluble sugar (Zhou et al., 2019; Dai et al., 2017), nucleic acids and protein (Chen & Cooper, 2002) were determined with the remaining supernatant.

To determine the total soluble protein content, we referred to the method of Bradford (1976), with some adjustments. Five agar discs of 1-F were transferred into 100 mL PDB and shaken at 180 rpm for 5 days at 28 °C. Then, the PE extract of GS-16 and mancozeb were added to the cultures at the EC50 concentration for incubation with shaking (180 rpm) for 2 days at 28 °C. Fresh mycelia were harvested, cleaned with sterile water, and blotted dry with sterile filter paper. Then, 0.5 g of the mycelia was added to 2 mL of phosphate-buffered saline (PBS) to be ground. The homogenate was centrifuged at 4 °C (10,000 × g, 20 min), and then, 1 mL of the supernatant was added to 5 mL of Coomassie brilliant blue G-250. The absorbance of the mixture was measured at 595 nm. Mancozeb was used as the positive control and sterile water was used as the negative control, and the experiments were performed in triplicate. The remaining fresh mycelia were used to determine the malondialdehyde (MDA) content (Fu & Huang, 2001).

Effects of the PE extract of GS-16 on 1-F cell metabolism

Five agar discs of 1-F were transferred to 100 mL of PDB medium and shaken at 180 rpm for 5 days at 28 °C. Fresh mycelia were harvested, rinsed with PBS and stained with 10 µmol/L 2,7-dichlorofluorescein diacetate (DCFH-DA) for 1 h at 37 °C (Han et al., 2015). Then, the mycelia were washed with PBS and placed in a 10 mL centrifuge tube. Finally, the PE extract of GS-16 and mancozeb were added to the mycelia at the EC50 concentration. Mancozeb was used as the positive control, and sterile water was used as the negative control. After the samples were allowed to stand for 10, 20, 30, 40 and 50 min, the fluorescence intensity was measured with an inverted fluorescence microscope (Olympus Co., Tokyo, Japan) and analyzed by ImageJ 17.0 software. The fluorescence intensity was determined as follows:

Fluorescenceintensity=FluorescenceintensityofthesampledareaSamplingarea×100

Fresh mycelia were collected as described in “Effects of the PE extract of GS-16 on the cell membrane permeability and cellular contents of 1-F”. Then, the succinate dehydrogenase (SDH) content and malate dehydrogenase (MDH) content in 1-F were determined by the SDH assay kit and MDH assay kit (Solarbao, Beijing, China), respectively, following the manufacturer’s instructions. The experiments were performed in triplicate.

Determination of defense enzyme activity

Two-year-old seedlings were randomly divided into four groups (three pots/group) and treated (spraying 5 mL/plant) as follows: (1) control group: sterile water; (2) pathogen treatment group: 1-F (1 × 106 spores/mL); (3) biocontrol treatment group: GS-16 (1 × 108 cfu/mL); and (4) pretreatment group: GS-16 (1 × 108 cfu/mL) treatment followed by 1-F application (1 × 106 spores/mL) 24 h later. The tea seedlings were placed in transparent plastic bags to prevent pathogen spread and watered with sterile water as needed. The seedlings were cultured on a 12 h light/dark photoperiod (temperature 25 °C, humidity 85%). The experimental treatments were repeated three times. The activities of five defense enzymes, polyphenol oxidase (PPO), superoxide dismutase (SOD), phenylalanine ammonia-lyase (PAL), peroxidase (POD), and catalase (CAT) were measured. After 1, 3, 5, 7, 9, and 11 days of incubation, 0.1 g of tea leaves was removed and ground with liquid nitrogen. The activities of PPO, SOD, PAL, POD, and CAT were determined by the PPO assay kit (Keming, Suzhou, China; PPO-2-W), SOD assay kit (Keming, Suzhou, China; SOD-1-W), PAL assay kit (Solarbio, Beijing, China; BC0090), POD assay kit (Solarbio, Beijing, China; BCO215), and CAT assay kit (Solarbio, Beijing, China; PB310), respectively, following the manufacturer’s instructions. The absorbance was measured by a UV‒VIS spectrophotometer (UV 752 N, YUANXI, Shanghai, China).

Data analysis

The data in this study are expressed as the mean ± standard deviation and were analyzed by one-way analysis of variance (ANOVA) using SPSS 24.0 (SPSS Inc., Chicago, Illinois, USA). Significant differences between treatment means were determined by Duncan’s multiple range test at the 5% level (p < 0.05). ImageJ 17.0 software was used to convert fluorescence intensity and gray values.

Results

Antagonistic activity and biocontrol efficacy of GS-16

In vitro and in vivo assays were used to assess for its antagonistic activity. Our result exhibited that GS-16 had stronger antifungal activity against 1-F, with an inhibition rate of 92.03%, and the BCF of GS-16 demonstrated an inhibition rate of 83.12% (Fig. 1 and Table 1). Moreover, GS-16 displayed different degrees of antifungal properties against eight different pathogens (Fig. 2 and Table 2), with the strongest inhibitory effects against F. graminearum Z-16 (74.54%), and F. equiseti B-3-1 was also susceptible to GS-16, with the weakest inhibition rate of 35.24%.

Figure 1 Antagonistic activities of GS-16 against 1-F in vitro.

(A) 1-F, (B) antagonistic effect of strain GS-16 against 1-F, (C) antagonistic effect of BCF against 1-F.

Table 1 Antagonistic activities of GS-16 against 1-F in vitro.

Treatment	Width of inhibition zone (cm)a	Inhibition rate (%)b	
Strain GS-16	0.70 ± 0.02b	92.03%	
BCF	1.48 ± 0.03a	83.12%	
Notes:

a Data are presented as the mean ± standard deviation (n = 3) and analyzed using Duncan’s Multiple Range Test. Different letters indicate significant differences (p < 0.05) within the same column.

b Data are shown for the mean of inhibition rate of triplicates.

Figure 2 Antagonistic test of GS-16 against eight pathogens on PDA.

CK: control cultures (no bacteria); GS-16: dual-culture of GS-16 against pathogens.

Table 2 Antagonistic test of GS-16 against 8 pathogens on PDA.

Plant pathogen	Disease	Width of
inhibition zone (cm)a	Inhibition rate (%)b	
Fusarium graminearum	Soybean root rot	2.20 ± 0.47	74.54a	
Fusarium
oxysporum	Banana fusarium wilt pathogen	2.45 ± 0.54	71.24a	
Colletotrichum camelliae	Brown blight disease	3.86 ± 0.16	46.45cd	
Alternaria alternate	Pear black spot pathogen	2.58 ± 0.06	64.19a,b	
Plectosphaerella cucumerina	Strawberry wilt	3.27 ± 0.22	56.46b,c	
Fusarium dimerum	Fusarium rot pathogen	2.54 ± 0.15	52.28b,c	
Sclerotium rolfsii	Pepper southern blight	4.15 ± 0.43	40.46d	
Fusarium equiseti	Soybean root rot	5.00 ± 0.34	35.24d	
Notes:

a Data are presented as the mean ± standard deviation (n = 3).

b Inhibition rate was analyzed using Duncan’s Multiple Range Test. Also, a, b, c and d indicate significant differences (p < 0.05) within the same column.

In the greenhouse experiment, GS-16 effectively attenuated 1-F infection of tea seedlings (Fig. 3 and Table 3). A total of 14 days after inoculation, the disease incidence with the 1-F spore suspension was 96.24%, while GS-16 reduced the disease incidence to 23.53%. There were no symptoms of tea anthracnose after treatment with sterile water. The disease index of seedlings treated with only the 1-F strain was 55.94. When treated with strain GS-16, the disease index decreased to 6.12, indicating that strain GS-16 has a good ability to control tea anthracnose (89.06%). Therefore, the results of the in vitro and in vivo assays showed that GS-16 has good antagonistic effects.

Figure 3 Biocontrol efficacy of GS-16 in the greenhouse.

Symptoms of tea plants treated with (A) sterile water, (B) 1-F, and (C) GS-16 and 1-F.

Table 3 Relative control effect of GS-16 on tea anthracnose disease in the greenhouse experiment.

Treatment	Disease incidence (%)a	Disease indexb	Relative control effect (%)c	
1-F	96.24 ± 0.12a	55.94 ± 1.10a		
GS-16+1-F	23.53 ± 0.70b	6.12 ± 0.38b	89.06%	
Notes:

a,b Data are presented as the mean ± standard deviation (n = 3) and analyzed using Duncan’s Multiple Range Test. Different letters indicate significant differences (p < 0.05) within the same column.

c Data are shown for the mean of relative control effect of triplicates.

EC50 of PE extracts quantification results

The results of the EC50 determination in this study were presented in Fig. 4. As shown in the figure, the PE extracts of GS-16 treatment had the strongest inhibition rate (76.20% growth inhibition) at 1.0 mg/mL, while the mancozeb showed the strongest inhibition rate (86.70% growth inhibition) at 0.5 mg/mL. The EC50 values of the mancozeb and PE extracts of GS-16 calculated by GraphPad Prism 9 software were 0.093 and 0.148 mg/mL, respectively.

Figure 4 EC50 of PE extracts quantification results.

Data are represented as the average of triplicates. Error bars represent the mean ± SD. Different letters above the columns indicate significant differences (p < 0.05).

SEM observations

To study the effect of PE extract of GS-16 on the mycelial morphology of 1-F, SEM was used to observe the mycelial morphology of 1-F. As shown in Fig. 5, the mycelia of 1-F maintained good growth and were regular in the negative group. In contrast, the PE extract of GS-16 caused deformities, distortions, and swelling in the mycelia of 1-F compared with the negative group. Some 1-F mycelia showed coarsening and creasing after treatment with mancozeb. Based on the results obtained, the PE extract of GS-16 may cause damage to the 1-F cell wall and further led to mycelial deformity.

Figure 5 SEM images of 1-F.

CK: untreated mycelial (negative control); mancozeb (positive control); PE extract of GS-16 treatment group. Bar = 40 μm.

Detection of extracellular enzymes

We evaluated whether the cell wall of 1-F was affected by GS-16, the extracellular enzyme activities of strain GS-16 were assessed by plate-based assays. The results in Fig. 6 showed that strain GS-16 was able to produce protease and cellulase, as indicated by the clear zones around the colony on CMC agar and skimmed milk agar medium, respectively (Fig. 6). The plate-based assays revealed that strain GS-16 could produce extracellular enzymes to destroy 1-F cell wall.

Figure 6 Detection of extracellular enzyme production.

(A) Cellulase, (B) protease.

Possible mechanism of GS-16 inhibiting 1-F

Effects of the PE extract of GS-16 on the cell membrane permeability and cellular contents of 1-F

The cell membrane plays a key role in cell growth. To evaluate the effects of the PE extract of GS-16 on the cell membrane of 1-F, the relative electrical conductivity, and contents of soluble sugar, nucleic acids, extracellular macromolecule, soluble proteins and MDA were measured.

As shown in Fig. 7A, the extracellular relative electrical conductivity in the GS-16 group was consistent with that of the positive control group. At 4 h, the PE extract of GS-16 significantly increased the relative electrical conductivity by 12.51% and 5.07% compared with the negative control and the positive control group, respectively (p < 0.05). The extracellular relative electrical conductivity in the GS-16 group and positive group exhibited less change from 8–32 h. These findings demonstrated that the PE extract affected the cell membrane permeability of 1-F, which led to electrolyte leakage into the extracellular environment.

Figure 7 Effects of PE extract of GS-16 on the cell membrane and cellular contents of 1-F.

(A) The relative electrical conductivity; (B) total extracellular soluble sugars; (C) nucleic acids leakage; (D) extracellular macromolecule content at 280 nm; (E) soluble protein content; (F) malondialdehyde (MDA) content. Data are represented as the average of triplicates. Error bars represent the mean ± SD. Different letters above the columns indicate significant differences (p < 0.05).

Next, the content of soluble sugar in the negative control group gradually decreased over time (Fig. 7B), and there was a significantly higher soluble sugar content in the GS-16 group than the negative control group at each detection time point, especially after treatment for 10 h (p < 0.05). When treated with the PE extract for 10 h, the soluble sugar content significantly increased by 79.26% in the GS-16 group compared to the negative control group, after which the soluble sugar content began to decrease but it still remained higher than that in the negative control group.

In addition, the nucleic acids leakage of 1-F caused by PE extract of GS-16 was presented in Fig. 7C. The PE extract of GS-16 treatment obviously increased the absorbance value (OD260) compared with that of the negative control and positive control at each detection time point. In particular, after treatment for 2 h, the GS-16 group displayed a significantly increased extracellular nucleic acids content (31.77%) compared with the negative control group (p < 0.05). Although the nucleic acids content in the GS-16 group began to decrease at 8 h, it was still higher than that of the positive control and the negative control groups. Meanwhile, the absorbance at 280 nm was higher in the GS-16 group than that in the negative control and peaked at 8 h (Fig. 7D), and the absorbance values of the GS-16 group and the positive control group were not significantly different. The absorbance values in the GS-16 treatment were 13.47% higher than those in the negative control group. These results indicated that treatment with the PE extract affected the cell membrane of 1-F, resulting in the extravasation of soluble sugar, intracellular nucleic acids and protein into the extracellular culture fluid.

Besides that, we further evaluated the effect of the PE extract on the normal synthesis of 1-F cellular contents. The soluble protein content in the GS-16 group was 25.14 ± 1.74 μg/mL (Fig. 7E). The result of Fig. 7E displayed that the soluble protein content in the GS-16 group was lower than in the negative control and the mancozeb control groups, and the difference among the three groups was significant (p < 0.05). These findings indicated that the PE extract severely damaged the cell membrane of 1-F, which caused the leakage of protein and nucleic acids, further affecting the synthesis of soluble proteins and finally resulting in cell death.

Figure 7F showed the MDA content in 1-F after treatment with the PE extract of GS-16. The MDA content in the GS-16 group was 0.24 ± 0.01 μmol/g FW, which was 61.84% that of the negative control group. There was no significant difference in MDA content between mycelia treated with PE extract or mancozeb. In conclusion, the results from the present study demonstrated that treatment with PE extract affected the 1-F cell membrane and augmented its permeability.

Effects of the PE extract of GS-16 on the cell metabolism of 1-F

We evaluated the degree of cell metabolism damage by determining 1-F after interaction with PE extract for the content of ROS. As shown in Fig. 8A, the ROS content of 1-F treated with the PE extract peaked at 5.53 ± 0.08 at 30 min, showing a significant increase of 31.23% compared with the negative control group (p < 0.05). At this time, there was no significant difference in ROS content between the GS-16 group and the mancozeb group (p > 0.05). In summary, the PE extract of strain GS-16 induced ROS accumulation and damaged cell metabolism in 1-F.

Figure 8 Cell metabolism change of 1-F after treatment with PE extract of GS-16.

The contents of (A) ROS, (B) MDH, and (C) SDH of 1-F treated with PE extract of GS-16. Data are represented as the average of triplicates. Error bars represent the mean ± SD. Different letters above the columns indicate significant differences (p < 0.05). Rosup used as reference.

The energy metabolism status of 1-F was assayed by detecting the key enzyme activities of SDH and MDH in the TCA cycle. As shown in Figs. 8B and 8C, the contents of MDH and SDH in 1-F treated with the PE extract of GS-16 were significantly lower than those in the negative control group, the MDH content decreased by 24.92% after treatment, while the SDH content decreased by 28.89% (p < 0.05). In addition, the results showed that there was no significant difference in the MDH content between mycelial cells treated with the PE extract and mancozeb (p > 0.05). These findings indicated that treatment with PE extract affected the TCA cycle in 1-F by inhibiting the activities of the key enzymes MDH and SDH in 1-F cells.

Induction of systemic resistance

The activities of SOD, CAT, PAL, PPO, and POD were examined to assess the effect of inducing host plant resistance by GS-16. The patterns of change among the five defense-related enzymes were presented in Fig. 9. In this study, the activities of SOD (Fig. 9A), CAT (Fig. 9B), PAL (Fig. 9C), PPO (Fig. 9D), and POD (Fig. 9E) reached the maximum at 5, 5, 7, 5, and 5 days in all treatments groups, respectively, and then subsequently declined. The SOD, CAT, PAL, PPO, and POD activities in the pretreatment group increased by 23.75%, 16.59%, 10.40%, 13.33%, and 14.24%, respectively, compared with those in the 1-F treatment group. The SOD, CAT, PAL, PPO, and POD activities in the biocontrol group increased by 31.17%, 22.30%, 12.79%, 18.47%, and 18.14%, respectively, compared with those in the control group (p < 0.05). Our data suggested that treatment with GS-16 stimulated the resistance response, resulting in enhanced defense-related enzyme activity.

Figure 9 Quantification of the plant defense-related enzyme activities.

CK: control group treated with sterile water; 1-F: pathogen treatment group; GS-16: biocontrol treatment group; GS-16+1-F: pretreatment group. (A) SOD, (B) CAT, (C) PAL, (D) PPO, (E) POD. Data are represented as the average of triplicates. Error bars represent the mean ± SD. Different letters above the columns indicate significant differences (p < 0.05).

Discussion

Anthracnose is a global disease of tea (C. sinensis) that causes severe damage and losses in tea plantations (Li, Zhang & Li, 2021). In recent years, biological control is a sustainable and eco-friendly measure to control tea anthracnose disease. Among the biological control options, endophytes have become an attractive resource for disease control in agricultural production (Latz et al., 2018). Tea plants have rich endophytes (Xie et al., 2020), and tea endophytes, such as Pseudomonas sp., Stenotrophomonas sp., Bacillus sp., and Lysinibacillus sp., have underlying PGP traits (Borah et al., 2019). B. altitudinis of Bacillus sp. has been classified as an endophytic bacterium, plant growth promoter and biological control agent (Ayilara, Adeleke & Babalola, 2022; Ye et al., 2022), for example B. altitudinis could significantly inhibit Streptomyces scabies in plate antagonism and pot experiments (Li et al., 2019), and B. altitudinis was obtained from the rhizosphere of Sechium edule, which exhibited a significant inhibition rate of 74% against Thanatephorus cucumeris (Sunar et al., 2015). Nevertheless, the biocontrol efficacy of B. altitudinis has been reported in only a few studies. Therefore, we sought to explore B. altitudinis for its potential in plant disease. This study revealed the antagonistic activity and biocontrol efficacy of the endophytic bacterium B. altitudinis GS-16 against C. gloeosporioides 1-F. Our in vitro showed that strain GS-16 significantly inhibited the growth of 1-F, and the inhibition rate of BCF was as high as 83.12%, indicating that GS-16 could produce active substances into the extracellular culture fluid. We should further analyze the secondary metabolites of GS-16 by GC–MS. Moreover, GS-16 exhibited significantly antagonistic activity against eight different pathogens. These results demonstrated that B. altitudinis GS-16 may have broad-spectrum antimicrobial properties (Yu et al., 2011). Biocontrol effect plays an important role in popularization and application of biocontrol agents. In pot experiments, the biocontrol efficacy of B. altitudinis GS-16 was 89.06%. We assumed that B. altitudinis could be used as an efficient biocontrol agent for tea anthracnose disease.

The most common mechanism of action of biocontrol bacteria against phytopathogens is the inhibition of mycelial growth (Jiang et al., 2018). As previously demonstrated, the BCF of Bacillus pumilus HR10 effectively weakened the mycelial growth of pathogens (Dai et al., 2021). The B. subtilis AF1 caused the lysis of fungal mycelia of Aspergillus niger (Podile & Prakash, 1996). In our study, the SEM results revealed that the PE extracts of GS-16 obviously suppressed the mycelial growth of 1-F and led to mycelial deformity, so we speculated that some antimicrobial substances was present in this PE extract (Fig. 5). The management of fungal phytopathogens with antimicrobial substances produced by Bacillus spp. is often one of the foremost biological control strategies (Fan et al., 2017). Bacillus spp. have been found to produce antimicrobial substances against phytopathogens, including lipid antibiotics and antimicrobial proteins (Zhao et al., 2017). B. subtilis EDR4 secretes antifungal proteins, thereby causing cell distortion in Gaeumannomyces graminis mycelia (Liu, 2008). In this study, our data showed that GS-16 could secrete extracellular enzymes, such as cellulose and protease, to inhibit the growth of 1-F (Fig. 6). Similarly, according to Potshangbam et al. (2018), B. altitudinis Lc5 secreted defensive enzymes like cellulase, β-1,3-glucanase, siderophore, protease, and chitinase, which was in agreement with the findings of this study. These results indicated that strain GS-16 could damage the cell walls of phytopathogens, and we believed that the extracellular enzymes produced by strain GS-16 were associated with the inhibition mechanism against phytopathogens.

The antifungal mechanism of the biocontrol microorganisms was also analyzed in the results of cell membrane integrity and permeability of phytopathogens. Biocontrol microorganisms could affect the physiological metabolism of the phytopathogens by altering the cell membrane permeability (Dai et al., 2021). The relative electrical conductivity indicated the level of cell membrane damage. In this study, the PE extract of GS-16 significantly enhanced the relative electrical conductivity of 1-F compared with the negative control group (Fig. 7A). These results were consistent with previous research findings, which showed that the PE extract of GS-16 could alter the cell membrane permeability of 1-F (Zhang et al., 2016). Thus, the cell membrane was damaged, causing the cellular components, including nucleic acids, proteins, and soluble sugar, to be released from the cell membrane. The cell membrane permeability and integrity were assessed by examining the leakage of cell contents into the supernatant at 260 and 280 nm (Mutlu-Ingok & Karbancioglu-Guler, 2017). The absorbance of the supernatant in the PE extract of GS-16 treatment group was stronger than that in the negative control group at 260 and 280 nm (Figs. 7C and 7D). These results showed that the PE extract could damage the integrity of the cell membrane. Moreover, the content of soluble sugar after GS-16 PE extract treatment was 79.26% higher than that in the negative control group at 10 h (p < 0.05), indicating that the PE extract of GS-16 damaged the cell membrane integrity, resulting in the extravasation of intracellular sugar substances into the extracellular culture fluid (Fig. 7B). The results showing the release of electrolyte, soluble sugar, nucleic acids and proteins were consistent with those from other some biological control agents treated on phytopathogens (Qian, Tao & Xie, 2010; Lee et al., 2014; Pan et al., 2022b). Furthermore, normal cellular metabolism is also related to the synthesis of cellular contents (Dai et al., 2021). Soluble proteins are the basic organic substances that act as osmotic regulators and are the main undertakers of life activities. In our study, the soluble protein content in the PE extract of GS-16 treatment group was significantly lower than that in the mancozeb group and the negative control group (p < 0.05) (Fig. 7E). The results of our study indicated that the PE extract of GS-16 could damage the cell membrane, affecting protein synthesis and increasing the leakage of soluble protein, which was in agreement with the results of Dai et al. (2021). Moreover, the MDA content indicates the level of stress-induced damage to the cell membranes (Toral et al., 2020). In this study, when the mycelia of 1-F were treated with the PE extract of GS-16, the MDA content increased significantly to 0.24 ± 0.01 μmol/g FW (Fig. 7F), which altered the cell membrane permeability. According to Cruz-Martín et al. (2018), B. pumilus CCIBP-C5 destroyed the cell membrane of Mycosphaerella fijiensis via a mechanism related to MDA accumulation, and our results were consistent with this finding.

Further analysis of the mode of action showed that the PE extract of GS-16 destroyed cellular metabolism in 1-F cells. ROS acts as an intracellular messenger, participating in energy metabolism and material exchange (Aguirre et al., 2005). However, excessive ROS accumulation causes oxidative stress, membrane damage and cell death (Takemoto, Tanaka & Scott, 2007; Khan et al., 2014). Therefore, the ROS content in 1-F cells was also evaluated. In this experiment, the ROS content in the PE extract of GS-16 group increased by 31.23% compared with that in the negative control group (p < 0.05) (Fig. 8A). These results indicated that the PE extract of GS-16 could induce oxidative stress to accumulate ROS, cause oxidative damage and cell death, which was consistent with the results of Hu et al. (2021). Based on the results shown by the antimicrobial effect of the PE extract of GS-16 on intracellular ROS in 1-F, the key enzyme activities of SDH and MDH in the TCA cycle in 1-F cell metabolism were determined. The TCA cycle is an epicenter for cell metabolism. Our study showed a significant decrease in the activities of the key enzymes MDH and SDH in the 1-F treated with the PE extract, which indicated that the PE extract of GS-16 could block the normal TCA cycle and affect normal cell metabolism (Figs. 8B and 8C). A recent study has found that the biocontrol microorganisms could inhibit normal cell metabolism in pathogens by effectively decreasing the activities of the enzymes MDH and SDH in the TCA cycle (Pan et al., 2022b).

In addition to the direct mechanisms of biocontrol microorganisms against pathogens, it is also well known that biocontrol microorganisms can induce plant resistance in the host plants. Often, the induction of plant resistance is related to defense enzymes, including PPO, chitinase, POD, β-1,3-glucanase and PAL (Karthikeyan et al., 2006; Rashad, Abdalla & Shehata, 2022). SOD is a primary plant antioxidant enzyme that is associated with the removal of superoxide radicals (Zhao et al., 2021). PAL can synthesize phenols, lignin, and other defense-related compounds (Tahsili et al., 2014; Gao, 2006), and PPO acts as a key enzyme that is associated with the oxidation of phenolic compounds and the synthesis of anthraquinones in passivating pathogens (Li et al., 2015). A study showed that Bacillus sp. can decrease the lesion area by inducing plant resistance (Prathuangwong & Buensanteai, 2007). B. subtilis has been shown to induce systemic resistance (ISR) associated with the synthesis of host enzymes, including POD, PPO, and SOD (Chowdappa et al., 2013). In this study, the activities of SOD, CAT, PAL, PPO, and POD in the pretreatment group were obviously higher than those in the 1-F treatment group, and the activities of the defense enzymes SOD, CAT, PAL, PPO, and POD in the biocontrol group were significantly higher than those in the untreated control group (p < 0.05), which verified that treatment with strain GS-16 increased the activities of these defense-related enzymes (Fig. 9). Strain GS-16 effectively decreased the lesion area associated with the induction of host plant resistance. These results are supported by previous studies implying that the application of some Bacillus strains to seedlings induces significantly systemic resistance in host plants (Kloepper, Ryu & Zhang, 2004; Szczech & Shoda, 2006).

Conclusions

In conclusion, our study explored a more effective endophytic bacterium for controlling tea anthracnose disease. This is the first report in which B. altitudinis, isolated from C. sinensis, was analyzed in terms of its antagonistic mechanisms against tea anthracnose disease. On the one hand, strain GS-16 showed strong inhibitory activity on 1-F and had broad-spectrum antagonistic ability in vitro. In a greenhouse experiment, strain GS-16 displayed good biocontrol efficacy against tea anthracnose disease. In addition, GS-16 secreted extracellular enzymes to destroy the cell wall of the pathogen and inhibited 1-F mycelial growth. Furthermore, GS-16 significantly altered the cell membrane permeability and damaged the integrity of 1-F, while destroying its cellular metabolism as well as its synthesis of cellular contents. On the other hand, strain GS-16 decreased the lesion area by inducing host plant resistance. In conclusion, GS-16 strongly inhibited tea anthracnose disease and could be an effective BCA against tea anthracnose disease.

Supplemental Information

Supplemental Information 1 Data on EC50 of GS-16, cell membrane permeability, cellular contents, cell metabolism, the induction of systemic resistance.

Click here for additional data file.

Additional Information and Declarations

Competing Interests

Author Contributions

Patent Disclosures

Data Availability

The authors declare that they have no competing interests.

Youzhen Wu conceived and designed the experiments, performed the experiments, analyzed the data, prepared figures and/or tables, and approved the final draft.

Yumei Tan conceived and designed the experiments, analyzed the data, prepared figures and/or tables, authored or reviewed drafts of the article, and approved the final draft.

Qiuju Peng conceived and designed the experiments, performed the experiments, analyzed the data, prepared figures and/or tables, and approved the final draft.

Yang Xiao conceived and designed the experiments, authored or reviewed drafts of the article, and approved the final draft.

Jiaofu Xie performed the experiments, prepared figures and/or tables, and approved the final draft.

Zhu Li conceived and designed the experiments, analyzed the data, authored or reviewed drafts of the article, and approved the final draft.

Haixia Ding conceived and designed the experiments, analyzed the data, authored or reviewed drafts of the article, and approved the final draft.

Hang Pan performed the experiments, authored or reviewed drafts of the article, and approved the final draft.

Longfeng Wei performed the experiments, authored or reviewed drafts of the article, and approved the final draft.

The following patent dependencies were disclosed by the authors:

Bacillus altitudinis GS-16 currently stored in the China General Microbiological Culture Collection Center (CGMCC, No. 24353).

The following information was supplied regarding data availability:

The raw measurements are available in the Supplemental Files.

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
