# Peer review of "Biocontrol potential of endophytic bacterium Bacillus altitudinis GS-16 against tea anthracnose caused by Colletotrichum gloeosporioides"

_PeerJ, doi:10.7717/peerj.16761_

## Round 0.1 · original submission · Major Revisions

This article need extensive improvement in language, methodology and description of results. I would recommend major revisions.

**Language Note:** The Academic Editor has identified that the English language must be improved. PeerJ can provide language editing services - please contact us at copyediting@peerj.com for pricing (be sure to provide your manuscript number and title). Alternatively, you should make your own arrangements to improve the language quality and provide details in your response letter. – PeerJ Staff

Reviewer 1 ·

Basic reporting

• The manuscript provides useful information to the audience. It is well written and structure with valid literature cited.
• Authors used spp. or species alternatively throughout the article such as Lines 55 and 67.
• It would be better to cite previous studies for Lines 75-76.
• Abbreviations should be used after the first time any term is mentioned in the writing such as Line 344.
• In introduction and methods sections, provide more details about why multiple isolates were used. It was first mentioned in Lines 88-93.
• Remove obviously in Line 223.

Experimental design

• Valid experimental design provided.

Validity of the findings

A few things to correct include are extra spacings such as Line 224, removing legends embedded in the sections, add more details to the figure legends so that it explains the complete story itself, use ‘It was previously found’ rather than ‘Li’ in Line 336.

Additional comments

NA

Reviewer 2 ·

Basic reporting

The manuscript by Wu et.al. reported the antifungal effect of bacterium GS-16, which could act as efficient biocontrol agents for managing tea anthracnose disease.

The authors have done an excellent work in Introduction and Methods sections, which provided nice background and purpose of the study, as well as detailed experimental procedures and conditions.

However, the main Results section was poorly written, which looked like a pile-up of figure legends without proper organization. I would recommend the authors re-write the Results section following the conventional structure with a brief statement of aim/purpose for each experiment before jumping into numbers, and followed by a brief summary/indication of the results.

In addition, since the effect of Bacillus species against tea anthracnose disease has been reported by multiple groups (Zhou et al. 2018; Shang et al 2021), could the authors add comments on the uniqueness or advantage of this study comparing to previous publications?

Experimental design

no comment

Validity of the findings

no comment

Reviewer 3 ·

Basic reporting

Your manuscript ¨Biocontrol potential of endophytic bacterium Bacillus altitudinis GS-16 against tea anthracnose caused by Colletotrichum gloeosporioides¨ is interesting but needs extensive revisions to be acceptable.
1) Firstly, I would recommend authors to read the manuscript critically and improve its English preferably it should be edited by a fluent English Speaker.
Abstract
1. Add more information. Just describing the isolated strain is insufficient.
2. Line 38: put and before the induction of systemic resistance.
3. Line 40: antagonistic activity (92%) against C. gloeosporioides and broad-spectrum antifungal activity; what do authors mean ¨ broad-spectrum antifungal activity¨.
Introduction
1. It is too short. I will recommend to expand by giving an over view of possible control measures. Provide strong justification to propose the current study.
Materials and Methods
Headings and subheadings should be revised e.g.Line 85, Test materials and test strains should be rephrased as ¨strains and conditions¨. Provide the reference of the strains and describe their characteristics briefly.
Pot experiment: It lacks necessary informations such as ¨How were the plants watered. How were spore adjusted (add detail), How were control (negative) separated from the other treatment to prevent aerial spread of fungi etc.
Results
1. There is need to describe all parameters in a comprehensive way. Some numeric figures have 2 digits after decimal while other have single. Need to rationalize and describe their significance in light of statistical outcome.
2. Avoid too much subheadings and merge the similar one.
Discussion
It lacks reasoning of the findings in context of the earlier reports. Highlight the novelty of the results.
Figures and Tables
All the figures should be labelled propoerly and add footnotes. Why the numeric data of some figures is not analyzed statistically?
Same for tables.

Experimental design

See Above

Validity of the findings

See Above

Additional comments

See Above

---

## Round 0.2 · Major Revisions

One reviewer has recommended acceptance however, still other two reviewers have major concerns about the data representation and language of the study. Authors should keenly revise the manuscript to meet the quality standard of PeerJ.

**Language Note:** The review process has identified that the English language must be improved. PeerJ can provide language editing services - please contact us at copyediting@peerj.com for pricing (be sure to provide your manuscript number and title). Alternatively, you should make your own arrangements to improve the language quality and provide details in your response letter. – PeerJ Staff

Reviewer 1 ·

Basic reporting

Provided.

Experimental design

Valid experimental design provided.

Validity of the findings

Figures/tables legends are still embedded in Results section.

Reviewer 2 ·

Basic reporting

Thanks the authors for addressing the comments well. The manuscript has been revised and improved as suggested.
No further comments.

Experimental design

No comments.

Validity of the findings

No comments.

Reviewer 3 ·

Basic reporting

Authors have conducted good experiments and the data generated is interesting. However, presentation of data is very poor. The English of text yet needs significant improvement. There are repetitive mistakes as indicated below (as an example)
Line 32: causing significant yield losses and great economic losses.
Line 36: antagonistic activity against tea anthracnose disease in our laboratory.
Why in lab, you have also conducted pot experiment as well.
Avoid single sentence paragraph. E.g.
Line 253-254: In the greenhouse experiment, GS-16 showed good biocontrol efficacy against tea anthracnose disease (Fig. 3 and Table 3).
Line 100: Never start sentence with numeric.
Some figs titles and footnotes also need to improve.
e.g. Fig 9: ¨Replace detection of antioxidants¨ with ¨Quantification of antioxidants¨

Experimental design

Ok

Validity of the findings

Ok

Additional comments

Authors have conducted good experiments and the data generated is interesting. However, presentation of data is very poor. The English of text yet needs significant improvement. There are repetitive mistakes as indicated below (as an example)
Line 32: causing significant yield losses and great economic losses.
Line 36: antagonistic activity against tea anthracnose disease in our laboratory.
Why in lab, you have also conducted pot experiment as well.
Avoid single sentence paragraph. E.g.
Line 253-254: In the greenhouse experiment, GS-16 showed good biocontrol efficacy against tea anthracnose disease (Fig. 3 and Table 3).
Line 100: Never start sentence with numeric.
Some figs titles and footnotes also need to improve.
e.g. Fig 9: ¨Replace detection of antioxidants¨ with ¨Quantification of antioxidants¨

---

## Round 0.3 · accepted · Accept

All suggestions have been incorporated.